# Clinical Application of Factor VIII:C to VWF:Ag Ratio for the Screening of Haemophilia A Carriers

**DOI:** 10.3390/jcm11061686

**Published:** 2022-03-18

**Authors:** Ki-Young Yoo, Soo-Young Jung, Jin-Young Choi, Hye-Ryeon Park, Young-Shil Park

**Affiliations:** 1Korea Hemophilia Foundation, Seoul 06641, Korea; gowho@hotmail.com (K.-Y.Y.); labyoug@naver.com (S.-Y.J.); cjy7002@hanmail.net (J.-Y.C.); n146857@empas.com (H.-R.P.); 2Department of Pediatrics, Kyung Hee University Hospital at Gangdong, Seoul 05278, Korea

**Keywords:** haemophilia, factor VIII, von Willebrand factor, carrier, screening

## Abstract

Analyses of factor VIII procoagulant activity (FVIII:C) and the FVIII:C to VWF:Ag ratio (FVIII:C/VWF:Ag ratio) have been investigated as screening bioassays to detect haemophilia carriers. This study aimed to determine the validity of the FVIII:C/VWF:Ag ratio and FVIII:C analyses as screening tests. We reviewed the medical records of 137 genetically confirmed, proband haemophilia A patients and 179 of their familial females who had undergone carrier testing. The collected data included the severity and mutation type of *F8* gene from probands and age, ABO blood type, FVIII:C, VWF:Ag, and the result of targeted gene analysis in females. We diagnosed 110 females as carriers, and their FVIII:C and FVIII:C/VWF:Ag ratio were lower than those in 69 non-carriers (FVIII:C: 59.3 IU/dL vs. 106.1 IU/dL, *p* = 0.000; FVIII:C/VWF:Ag ratio: 0.62 vs. 1.08, *p* = 0.000). In receiver operating characteristic analysis, the areas under the curve (AUC) of the FVIII:C/VWF:Ag ratio and FVIII:C were 0.936 and 0.876, respectively. The cut-off value of FVIII:C/VWF:Ag ratio (0.81) at the maximum Youden J index provided a sensitivity of 82.8% and specificity of 96.6%. The cut-off value of FVIII:C (83.8 IU/dL) showed a sensitivity of 81.8% and specificity of 79.7%. Considering the AUC, the FVIII:C/VWF:Ag ratio is a good screening test to detect haemophilia A carriers, as evidenced by its specificity of 96.6%; however, it may also induce false-negative results.

## 1. Introduction

Congenital haemophilia A is one of the most common congenital bleeding disorders. Almost all of the patients are male because X-linked recessive traits transmit hereditary haemophilia A. However, haemophilia A may also develop sporadically. Mild to moderate haemophilia A is familial in 70% of patients, whereas severe haemophilia A is familial in only 45% of patients [1]. In Korea, 55.7% of patients with haemophilia A are familial [2]. Congenital haemophilia A affects 1 in 5000–10,000 male births, and coagulation factor VIII encoded by the *F8* gene is deficient in haemophilia A. In X-linked inheritance, a female heterozygote with a mutation on only one of the two X-chromosomes, the so-called carrier, often has FVIII:C of more than 50 IU/dL of normal activity. Approximately two-thirds of carriers have FVIII:C > 60 IU/dL [3]. Because a substantial number of carriers have normal FVIII:C and are symptom-free, it is challenging to diagnose carriership based on FVIII:C or bleeding symptoms. Rizza et al. reported that they could distinguish only 35% of carriers based on the FVIII:C value alone [4].

To diagnose a carrier, researchers have applied direct and indirect mutation analysis of the *F8* gene. Linkage analysis using various restriction fragment length polymorphisms (RFLPs) or variable numbers of tandem repeats is representative indirect mutation analysis. Although linkage analysis is a technically simple and cost-saving method, the misdiagnosis rate attributable to DNA recombination error ranges from 1% to 5% depending on restriction enzyme markers [5,6]. The interpretation of linkage analysis may face limitations in sporadic haemophilia A, but linkage analysis still has its role to play, especially in families where direct mutations have not been found.

For these reasons, direct mutation analysis, such as intron 22 inversion or DNA sequencing, is preferred even though it is labour-intensive and costly. Pathogenic mutations can be identified in 98% of patients by direct mutation analysis [7]. On the other hand, when neither linkage analysis nor direct mutation analysis is available for any reason, several researchers have studied coagulation factor parameters such as FVIII:C/VWF:Ag ratio or FVIII:C to screen haemophilia A carriership [4,8,9,10,11]. However, to the best of our knowledge, pathogenic mutations have never been adopted as references for these studies. Here, we used pathogenic mutations from proband haemophilia A patients as a reference compared to coagulation factor testing for the screening of carriership. This study aimed to determine the validity of the optimal cut-off value of FVIII:C/VWF:Ag ratio and to compare the FVIII:C/VWF:Ag ratio and FVIII:C value as screening tests for haemophilia A carriers.

## 2. Materials and Methods

### 2.1. Study Population

We retrospectively reviewed the medical records of 137 patients genetically confirmed with haemophilia A and 179 of their familial females who had undergone carrier testing between April 2007 and May 2021 at the Korea Hemophilia Foundation (KHF) clinic. For the avoidance of bias which could be caused by very low FVIII:C, female haemophilia A patients having less than 5 IU/dL of FVIII:C and typical bleeding diathesis such as haemarthrosis were excluded. We collected information on the disease severity and mutation type of F8 gene from probands. From the subjected females, age, family relationship with probands, bleeding diathesis, ABO blood type, FVIII:C, VWF:Ag, and the results of targeted gene analysis were derived. We collected the information regarding bleeding diathesis from verbal questions and answers.

### 2.2. Coagulation Assay

FVIII:C was measured using the one-stage assay with ACL 9000TM (Werfenmedical IL, Bedford, MA, USA) using Synthasil^®^ reagent or CS-2500 automated coagulation analyser (Sysmex Corporation, Kobe, Japan) using the Dade Actin FS (Siemens Healthcare Diagnostics, Marburg, Germany) reagent. VWF:Ag was measured with an ACL 9000TM automated coagulation analyser using HemosILTM VWF antigen (Werfenmedical IL, Bedford, MA, USA).

### 2.3. Mutation Analysis

The gold standard for the identification of a female as a haemophilia carrier was genetic testing. Inversion testing, DNA sequencing and the gene dosage assay were subsequently carried out until the detection of the pathogenic mutations. Long-distance PCR or inverse-shifting PCR according to the research of Rossetti et al. was utilized for inversion tests [12,13]. Sanger sequencing with ABI 3130 Genetic Analyser (Applied Biosystems, Waltham, MA, USA) or next-generation sequencing using MiSeqDx^®^ (Illumina Inc., San Diego, CA, USA) was performed for DNA sequencing. Multiple ligation-dependent probe amplification was applied for the gene dosage assay. As for females who had wanted to undergo carrier testing, we carried out targeted gene mutation analysis for the same pathogenic mutation as their own probands. We classified the pathogenic mutations into null (inversion, nonsense, and frameshift) and non-null mutations (missense and splicing).

### 2.4. Statistical Analysis

Statistical analysis was performed using SPSS Statistics version 20 (IBM Corp., Armonk, NY, USA). Data that satisfied the normality requirement were expressed as the mean ± SD. Otherwise, the data are described as median (range). Differences between carrier subjects and non-carrier subjects were analysed using the chi-square test and Mann–Whitney U test depending on the variable scales. Linear correlation analysis was used to determine the correlation between FVIII:C and VWF:Ag. In addition, binomial logistic regression was used for correlation analysis between FVIII:C and bleeding diathesis. Statistical significance was defined as a *p*-value < 0.05. We performed ROC analysis using MEDCALC^®^ (MedCalc Software Ltd., Ostend, Belgium; https://www.medcalc.org/ (accessed on 19 July 2021)) software. The target sample size was calculated as 143 using G*power 3.1.9.7, with an effect size of 0.3, a power of 0.8, and a significance level of 0.05 [14]. The optimal cut-off was determined by the criterion value, which was automatically calculated at the maximum Youden J index [15].

## 3. Results

### 3.1. Characteristics of the Study Population

A total of 316 subjects were included in this study (Table 1). A single female from each family of 105 probands underwent carrier testing, while a variable, plural number of females from each family of the remaining 32 probands carried out the test. This made the number of females larger than that of probands in this study. Severe probands accounted for 112 of 137 (81.8%), and pathogenic mutations could be identified in 61.5% (110 out of 179) of females diagnosed as carriers. No pathogenic mutations were identified in the remaining 69 females diagnosed as non-carriers. Interestingly, the proportion of carriers among mothers was 92.7%, whereas that of carriers and non-carriers among siblings and other family members was even. The number of females with O and non-O blood types was 48 and 100, respectively. Blood type A was the most frequent (33.1%), followed by the O blood type (32.4%). The ABO blood type of 31 females was unidentified. There was no statistical difference in carriership between the O blood type and non-O blood type females. Regarding mutation types of probands, null mutations accounted for 61.5% (110/179). The most common mutation was missense, which was identified in 34.1%, and intron 22 inversion was detected in 32.4%. No significant differences in mutation types were observed between carriers and non-carriers.

### 3.2. Bleeding Diathesis

Bleeding diathesis was investigated in the medical records of 74.9% (134/179) of the females. While 68 females reported not increased bleeding tendency, 66 females complained of increased bleeding manifestations. Multiple bleeding diathesis was identified in 12 out of 45 symptomatic carriers and 2 out of 21 symptomatic non-carriers. The most frequent bleeding diathesis was hypermenorrhoea, followed by subcutaneous haematoma and epistaxis. The difference was statistically insignificant (*p* = 0.052), but carriers reported bleeding diathesis more commonly (55.5%, 45/81) than non-carriers (39.6%, 21/53), especially subcutaneous haematoma, epistaxis and traumatic bleeding.

### 3.3. FVIII:C, VWF:Ag, and FVIII:C/VWF:Ag Ratio

FVIII:C was measured in all females, but VWF:Ag was tested in 160 females. The median FVIII:C, VWF:Ag, and FVIII:C/VWF:Ag ratio of the total females were 74.5 IU/dL, 103.0 IU/dL, and 0.77 respectively (Table 2). In carriers, the FVIII:C, VWF:Ag, and FVIII:C/VWF:Ag ratio were 59.3 IU/dL, 101.0 IU/dL, and 0.62, respectively. Corresponding figures of each parameter in non-carriers were 106.1 IU/dL, 105.0 IU/dL and 1.08. Both FVIII:C and FVIII:C/VWF:Ag ratios were significantly lower in carriers than in non-carriers (*p* = 0.000). These findings were similar when the data of carriers and non-carriers defined as O or non-O blood type subjects were compared. FVIII:C and VWF:Ag were lower in the O blood type than in the lower non-O blood type (*p* = 0.000). However, the FVIII:C/VWF:Ag ratio was not significantly different (O blood type, 0.73; non-O blood type, 0.77; *p* = 0.770) (Figure 1). Between asymptomatic and symptomatic females, no differences in FVIII:C (asymptomatic: 77.8 IU/dL vs. symptomatic: 68.6 IU/dL, *p* = 0.976), VWF:Ag (asymptomatic: 102.0 IU/dL vs. symptomatic: 92.2 IU/dL, *p* = 0.387), and FVIII:C/VWF:Ag ratio (asymptomatic 0.83 vs. symptomatic 0.77, *p* = 0.620) were observed. There was a strong positive correlation between FVIII:C and VWF:Ag (r = 0.459, *p* = 0.000), and the correlation was stronger in non-carriers (r = 0.756, *p* = 0.000) than in carriers (r = 0.483, *p* = 0.000). In terms of mutation types, there was no statistical difference in the FVIII:C, VWF:Ag, and FVIII:C/VWF:Ag ratio. There was no significant association between FVIII:C and bleeding diathesis (odds ratio, 1.003; 95% CI, 0.994–1.011; *p* = 0.515). However, females with FVIII:C < 40 IU/dL manifested an increased bleeding tendency (*p* = 0.045) compared to females with FVIII:C > 40 IU/dL. When it was defined to carriers, bleeding diathesis became more remarkable in carriers with FVIII:C of less than 40 IU/dL (*p* = 0.004). On the other hand, 22 carriers and none of the non-carriers had FVIII:C less than 40 IU/dL.

### 3.4. Validity of FVIII:C/VWF:Ag Ratio and FVIII:C

Using MEDCALC^®^, the sensitivity and specificity of FVIII:C/VWF:Ag ratio at criterion value of 0.73 were estimated at 75.8% and 96.6%, respectively. Given that the prevalence of haemophilia A carriage was 0.02%, the positive and negative predictive values were 0.86% and 99.99%, respectively. We performed ROC analysis to determine the optimal cut-off value (Figure 2). The criterion value of the FVIII:C/VWF:Ag ratio at a maximum Youden J index of 0.79, which corresponds to the optimal cut-off value, was 0.81. At this point, the sensitivity and specificity were 82.8% and 96.6%, respectively. It is important to note that the AUC of FVIII:C/VWF:Ag was 0.936.

In terms of FVIII:C, the cut-off value was determined at 83.8 IU/dL with a Youden J index of 0.62. The AUC (0.876), sensitivity (81.8%), and specificity (79.7%) were lower than those of FVIII:C/VWF:Ag.

## 4. Discussion

Authors should discuss the results and how they can be interpreted from the perspective of previous studies and of the working hypotheses. The findings and their implications should be discussed in the broadest context possible. Future research directions may also be highlighted.

*F8* mutation analysis is the gold standard for diagnosing haemophilia A carriers. Nevertheless, several studies regarding FVIII:C/VWF:Ag ratio and FVIII:C were conducted to determine the possibility of screening tests for haemophilia A carriership [4,8,9,10,11]. In these studies, various methods such as RFLP, pedigree analysis, and direct mutation analysis have been used solitarily or jointly to confirm carriership. However, rates of misdiagnosis inherent to intragenic and extragenic restriction enzyme markers for RFLP are approximately 1% and 3–5%, respectively. In addition, the pedigree analysis may sometimes be incorrect or confabulated. It should be noted that this study used direct mutation analysis alone as a reference test to avoid misdiagnosis of RFLP and incorrect pedigree analysis.

The proportion of severe probands in this study was approximately 10% higher than that of the KHF registrants [16]. It is assumed that, in comparison with mild to moderate patients, severe patients experience more frequent bleeding and family members bear a heavier disease burden. Hence, females among the family members of severe patients might feel the necessity of carrier testing more strongly.

Bleeding diathesis in carriers has been reported in a variable range. While Paroskie et al. reported that 72% of obligate haemophilia A carriers had a high frequency of bleeding symptoms [17], Seck et al. reported that only 18.1% of carriers presented with bleeding symptoms [18]. In our study, bleeding diathesis was reported in 57.3% of carriers and 40% of non-carriers. The reason why bleeding diathesis was relatively high in both carriers and non-carriers is unclear. A possible explanation is the preconceived knowledge of non-carriers about haemophilia bleeding.

The correlation between bleeding diathesis and FVIII:C is also controversial. Seck et al. reported the FVIII:C level was significantly different between carriers with and without bleeding diathesis (42 ± 8.61 IU/dL vs. 100 ± 50.95 IU/dL, *p* = 0.001) [18]. However, Olsson et al. demonstrated an insignificant or weak negative correlation (rs = −0.36, *p* < 0.001) between FVIII:C and bleeding diathesis [19]. This finding is similar to ours: our study found no significant correlation between FVIII:C and bleeding diathesis. Even though bleeding diathesis was more frequently reported in carriers with FVIII:C less than 40 IU/dL, the number was too small to draw a conclusion.

VWF represents a high-molecular weight adhesive glycoprotein that plays an essential role in primary haemostasis by promoting platelet adhesion to the subendothelium and platelet plug formation at the sites of vascular injury. VWF has a second role in haemostasis which is to bind FVIII and protect it from premature clearance and degradation [20,21,22]. It is known that non-O blood type individuals have higher (approximately 25%) FVIII:C and VWF:Ag than O blood type individuals [23,24]. A recently published single-centre study found that individuals with blood group O had approximately 10–20% lower plasma VWF levels compared to individuals without blood group O [25]. It has been suggested that the lower VWF:Ag in O blood type individuals is attributable to the susceptibility of VWF to ADAMTS13 proteolysis and the significantly increased hepatic clearance of VWF, which is mediated via the asialoglycoprotein receptor. ABO antigens do not influence the rate of VWF synthesis or secretion [26]. The effect of the O blood type on the FVIII:C level was less than that on VWF:Ag [19]. In a study of monozygotic and dizygotic twin pairs, no effect of ABO blood type on FVIII:Ag remained after the correction of FVIII levels for VWF values (*p* = 0.62). This result indicated that the significant impact of ABO type on FVIII:C level was mediated through an effect on VWF:Ag. In our study, 31.1% of VWF:Ag and 33.2% of FVIII:C were reduced in O blood type subjects compared to those of non-O blood type subjects. A stronger correlation of FVIII:C and VWF:Ag in non-carriers (r = 0.781) than in carriers (r = 0.556) suggests that FVIII:C levels are more dependent on VWF:Ag levels in non-carriers.

For the screening of diseases with a heavy burden, a test with high sensitivity is usually preferred, and it is known that a measure with high specificity is more eligible for the screening of curable or treatable diseases. Shetty et al. indicated that the lowest misclassification rate of 7% among the carriers was seen when a cut-off value of 0.7 was chosen [9]. The Youden J index is most commonly used because it suggests a consistent cut-off value irrespective of prevalence [27]. The Youden J index determines the optimal cut-off at the point where the difference between the true positive rate and false positive rate is maximal among all ROC curve points. According to the Youden J index, the criterion value of the FVIII:C/VWF:Ag ratio was determined to be 0.81, which corresponds to the optimal cut-off value. The sensitivity and specificity at the cut-off value were 82.3% and 96.6%, respectively. Considering this low sensitivity and high specificity, a person who has a positive result can be strongly suspected as a carrier. However, false-negative interpretations should be considered when a subject obtains a negative result.

Even though the AUC of FVIII:C showed very good results, the AUC of the FVIII:C/VWF:Ag ratio was excellent; thus, the FVIII:C/VWF:Ag ratio rather than FVIII:C should be considered for the screening of haemophilia A carriers [28].

This study has some limitations. First, data on bleeding diathesis were collected using simple verbal questions and answers. Bleeding assessment tools, such as quantitative bleeding score, might be essential to conclude bleeding diathesis in carriers. Second, this study did not consider the menstrual cycle and oral contraceptives which may affect the FVIII:C and VWF:Ag levels.

## 5. Conclusions

In conclusion, the gold standard for the diagnosis of haemophilia A carriership is mutation analysis. However, in case genetic analysis of the pathogenic mutation is unavailable, the FVIII:C/VWF:Ag ratio can be considered as at least a screening test.

## Figures and Tables

**Figure 1 jcm-11-01686-f001:**
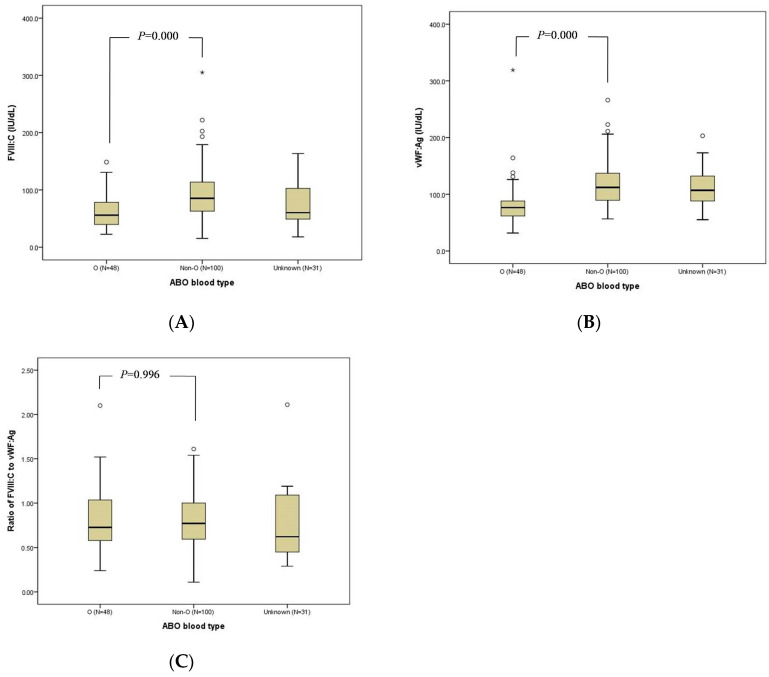
FVIII:C (**A**), VWF:Ag (**B**), and FVIII:C/VWF:Ag ratio (**C**) between O and non-O blood type.

**Figure 2 jcm-11-01686-f002:**
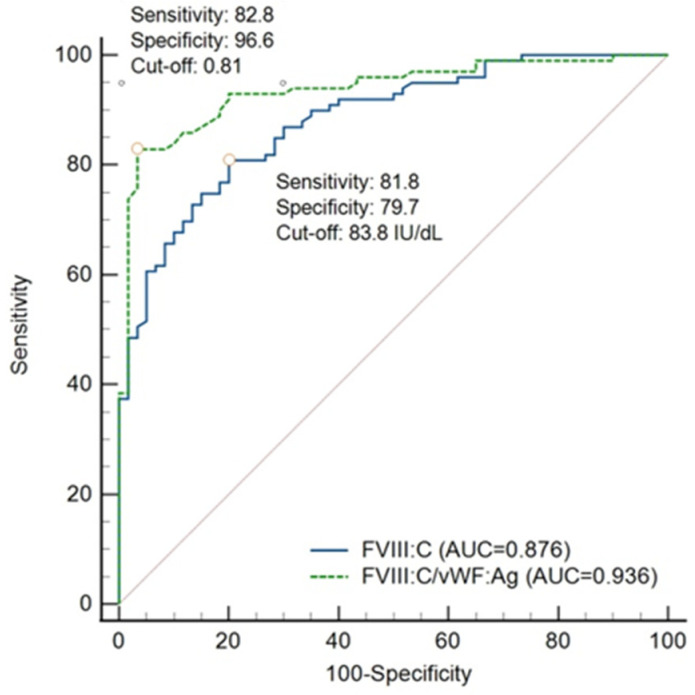
Receiver operating characteristics analysis of FVIII:C/VWF:Ag ratio and FVIII:C. Abbreviations: AUC, area under curve.

**Table 1 jcm-11-01686-t001:** Characteristics of subjected females.

		Overall (*n* = 179)	Carrier (*n* = 110)	Non-Carrier (*n* = 69)	*p*
Median (range) age, years		31 (3–68)	32 (5–68)	28 (3–67)	0.003
Relationship with proband, *n* (%)	Mother	55	51 (92.7)	4 (7.3)	
	Sibling	78	35 (44.9)	43 (55.1)	
	Ascendant or descendant	46	24 (52.2)	22 (47.8)	
ABO Blood type, *n* (%)	O	48	33 (68.8)	15 (31.2)	0.138
	Non-O	100	56 (56.0)	44 (44.0)	
	Unknown	31	21 (67.7)	10 (32.3)	
*F8* mutation types of proband, *n* (%)	Intron 22 inversion	58	38 (65.5)	20 (34.5)	0.489
	Missense	61	35 (57.4)	26 (42.6)	
	Nonsense	25	15 (60.0)	10 (40.0)	
	Frameshift	24	18 (75.0)	6 (25.0)	
	Splicing	8	2 (25.0)	6 (75.0)	
	Intron 1 inversion	3	2 (66.7)	1 (33.3)	
Bleeding diathesis, *n* (%)	Asymptomatic	68	36 (52.9)	32 (47.1)	0.052
	Hypermenorrhoea	35	22 (62.9)	13 (37.1)	
	Subcutaneous Haematoma	26	18 (69.2)	8 (30.8)	
	Epistaxis	12	11 (91.7)	1 (8.3)	
	Traumatic bleeding	7	7 (100.0)	0	
	Gum bleeding	1	1 (100.0)	0	
	Multiple diathesis	14	12 (85.7)	2 (14.3)	
	Unknown	45	29 (64.4)	16 (33.6)	

**Table 2 jcm-11-01686-t002:** FVIII:C, VWF:Ag, and FVIII:C/VWF:Ag ratio by carriership and ABO blood type.

Variables	ABO Blood Type	Total	Carrier	Non-Carrier	*p* *
Median FVIII:C, IU/dL, (*n*, range)	Overall	74.5 (179, 15.4–305.0)	59.3 (110, 15.4–130.6)	106.1 (69, 50.0–305.0)	0.000
	O	56.0 (48, 22.6–148.5)	47.5 (33, 22.6–130.6)	86.4 (15, 52.0–148.5)	0.000
	Non-O	85.3 (100, 15.4–305.0)	67.0 (56, 15.4–126.2)	112.6 (44, 65.8–305.0)	0.000
	Unknown	60.3 (31, 18.0–163.4)	54.3 (21, 18.0–126.0)	112.1 (10, 50–163.4)	NA
Median VWF:Ag, IU/dL, (*n*, range)	Overall	103.0 (160, 31.7–319.0)	101.0 (99, 31.7–319.0)	105.0 (61, 41.2–223.0)	0.671
	O	76.6 (48, 31.7–319.0)	77.2 (33, 317–319.0)	69.6 (15, 41.2–138.0)	0.468
	Non-O	112.0 (95, 56.6–266.0)	113.0 (54, 56.6–266.0)	110.6 (41, 70.8–223.0)	0.887
	Unknown	107.0 (17, 55.2–203.0)	106.5 (12, 55.2–203.0)	116.0 (5, 65.3–124.0)	NA
Median FVIII:C/VWF:Ag, (*n*, range)	Overall	0.77 (160, 0.11–2.11)	0.62 (99, 0.11–1.43)	1.08 (61, 0.57–2.11)	0.000
	O	0.73 (48, 0.24–2.10)	0.62 (33, 0.24–1.19)	1.09 (15, 0.87–2.10)	0.000
	Non-O	0.77 (95, 0.11–1.61)	0.63 (54, 0.11–1.43)	1.06 (41, 0.57–1.61)	0.000
	Unknown	0.62 (17, 0.29–2.11)	0.48 (12, 0.2—1.19)	1.12 (5, 1.02–2.11)	NA

* *p* value between carriers and non-carriers. Abbreviations: NA, not applicable.

## Data Availability

Data are available upon reasonable request from the corresponding author.

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
