# Peer review of "Clinical Application of Factor VIII:C to VWF:Ag Ratio for the Screening of Haemophilia A Carriers"

_jcm, 2022, doi:10.3390/jcm11061686_

Round 1
Reviewer 1 Report
The authors of the manuscript focused on clinical use of the FVIII/vWF ratio for the diagnosis of haemophilia A carriers. FVIII/VWF:Ag ratio is better than FVIII level alone in identifying carriers but it is not 100% sensitive even in women with a high a-priori likelihood of being carriers. Genetic testing therefore remains the gold standard for carrier determination and should be pursued to correctly label women as carriers/non-carriers.
The methodological part included a retrospective analysis 137 patietns with hemophilia A and 179 femals who had undergone carrier in Korea.
In the last paragraph in the introduction, the authors should describe vWF and FVIII in detail. VWF represents a high-molecular-weight adhesive glycoprotein that plays an essential role in the primary hemostasis by promoting platelet adhesion to the subendothelium and platelet plug formation at the sites of vascular injury. VWF also stabilizes and protects factor VIII in the circulation. At the same time, they should cite two recent publications in which these data have been presented. ,,Semin Thromb Hemost 2017; 43(06): 639-641 DOI: 10.1055/s-0037-1603362 and Blood Transfus. 2016 May;14(2):262-76. doi: 10.2450/2016.0258-15“.
Page 7 In addition, VWF levels in individuals also differ on a genetic basis, with the most common effect being found in the ABO blood group—individuals with blood group O have about 10–20% lower plasma VWF levels. These important facts should be included by the authors in this section and at the same time to cite the manuscript of a monocentric study in patients with von Willebran disease. ,,Diagnostics (Basel). 2021 Nov 20;11(11):2153. doi: 10.3390/diagnostics11112153.“
I have to say that with these 24 references but 4 references are from last 5 years, authors should add newer references
Author Response
I appreciate your sincere and precise comments. I revised and checked my article according to your comments.
1. In the last paragraph in the introduction, the authors should describe vWF and FVIII in detail. VWF represents a high-molecular-weight adhesive glycoprotein that plays an essential role in the primary hemostasis by promoting platelet adhesion to the subendothelium and platelet
plug formation at the sites of vascular injury. VWF also stabilizes and protects factor VIII in the circulation. At the same time, they should cite two recent publications in which these data have been presented. ,,Semin Thromb Hemost 2017; 43(06): 639-641 DOI: 10.1055/s-0037-1603362 and Blood Transfus. 2016 May;14(2):262-76. doi: 10.2450/2016.0258-15“.
→ Thank you for the detailed point. I added and revised the relevant content and recommended references to the review section. It was thought that it would be more appropriate to add to the discussion rather than the introduction described as detection of haemophilia A carrier.
2. Page 7 In addition, VWF levels in individuals also differ on a genetic basis, with the most common effect being found in the ABO blood group—individuals with blood group O have about 10–20% lower plasma VWF levels. These important facts should be included by the authors in this section and at the same time to cite the manuscript of a monocentric study in
patients with von Willebran disease. ,,Diagnostics (Basel). 2021 Nov 20;11(11):2153. doi:10.3390/diagnostics11112153.“
→I added related sentences to the relevant parts of the review, and also inserted references.
3. I have to say that with these 24 references but 4 references are from last 5 years, authors should add newer references
→I added recent reference according to your suggestion.
Reviewer 2 Report
Although revisiting a well trodden path this work nicely balances the findings of the first studies on carrier detectin by phenotype with the currnet gold standard of mutation specific diagnosis. The earlier approach is still valid as there are many families where either the mutation is not known because of lack of a proband or as in up to 5% of cases no mutation can be detected even in the proband. Linkage analysis still has its role to play in such circumstances.
I detected one minor confusing statement in the last sentence of the second pragraph on page three. What is meant is that the type of mutation did not differ between families where the consultand was either a carrier or not a carrier. Perhaps a bit more could be said about the role of linkage analysis in difficult kindered where no specific mutation has been found, but they did not include such families in this analysis, or if they did it was not mentioned. The observations about the degree of correlation of factor VIII with von Willebrand factor levels being higher in severe mutations than mild is intriguing. The correlation of bleeding diathesis with carriership was concluded to be non-significant but inspection of the individual bleeding types show highly significant correlation with traumatic bleeding, epistaxis, deep bruising and multiple types. Hence I would conclude that there is a correlation and that it needs to be analysed in a different way, such as a standarised bleeding score and expert evaluation.
Author Response
I appreciate your sincere and precise comments. I revised and checked my article according to your comments.
1. The earlier approach is still valid as there are many families where either the mutation is not known because of lack of a proband or as in up to 5% of cases no mutation can be detected even in the proband. Linkage analysis still has its role to play in such circumstances.
→ I revised the relevant content. I added this sentences in the introduction, “linkage analysis still has its role to play, especially in families where direct mutations have not been found”.
2. I detected one minor confusing statement in the last sentence of the second pragraph on page three. What is meant is that the type of mutation did not differ between families where the consult and was either a carrier or not a carrier. Perhaps a bit more could be said about the role of linkage analysis in difficult kindred where no specific mutation has been found, but
they did not include such families in this analysis, or if they did it was not mentioned.
→ Thank you for the sharp point. For the avoidance of confusion, I have mentioned them in the introduction, as previously mentioned. The purpose of this paper was to validate the cut-off value of the FVIII/VWF ratio by taking only the proband from which the direct mutation result was obtained as a reference. Of course, there were kindreds that performed linkage analysis because direct
mutations did not appear in the proband during the study period, but such families were excluded because they did not meet the purpose of the study.
3. The correlation of bleeding diathesis with carriership was concluded to be non-significant but inspection of the individual bleeding types show highly significant correlation with traumatic bleeding, epistaxis, deep bruising and multiple types. Hence I would conclude that there is a correlation and that it needs to be analysed in a different way, such as a standarised bleeding
score and expert evaluation.
→ Thank you for the detailed point. The paper has been revised as follows. The difference was statistically insignificant (p = 0.052), but carriers reported bleeding diathesis more commonly (55.5%, 45/81) than non-carriers (39.6%, 21/53), especially subcutaneous hematoma, epistaxis and traumatic
bleeding.
Round 2
Reviewer 1 Report
The presented manuscript has been corrected in response to the suggestions. The authors have followed the recommendations of the reviewer. After the revision, the provided data and addition of the results became more clear. I would like to thank the authors for resubmitting the manuscript and explaining the obscure points from the previous version.